Elevated albumin-bilirubin score as a predictor of kidney stones in adults with type 2 diabetes mellitus: evidence from a cross-sectional study

Mo Wenya 1
Zhou Tao 1 2
Dong Qifei 1
Chen Yuhan 1
Hu Xuechun 1
Xiao Jun 1
Wang Changming anhui930602wcm@126.com 1
Xiang Ping xiangping25@163.com 1 2
1 Department of Urology, The First Affiliated Hospital of USTC, Division of Life Sciences and Medicine, University of Science and Technology of China , Hefei , China
2 Graduate School, Bengbu Medical University , Bengbu , China
Capusa Cristina
Electronic publication date: 2025 May 14
Publication date: 2025
Volume: 13
Electronic Location ID: e19419
Received 2024 Dec 10; Accepted 2025 Apr 14
Copyright: ©2025 Mo et al.
Copyright year: 2025
Copyright holder: Mo et al.
License: This is an open access article distributed under the terms of the Creative Commons Attribution License, which permits unrestricted use, distribution, reproduction and adaptation in any medium and for any purpose provided that it is properly attributed. For attribution, the original author(s), title, publication source (PeerJ) and either DOI or URL of the article must be cited.
License URL: https://creativecommons.org/licenses/by/4.0/

Keywords: ALBI score, Kidney stones, Type 2 diabetes mellitus, Cross-sectional study

Funding: Anhui Provincial Key Clinical Specialties Construction Project (2023) Scientific Research Project of Universities of the Department of Education of Anhui Province No. 2022AH051256 The funding support was provided by Anhui Provincial Key Clinical Specialties Construction Project (2023) and the Scientific Research Project of Universities of the Department of Education of Anhui Province (No. 2022AH051256). The funders had no role in study design, data collection and analysis, decision to publish, or preparation of the manuscript.

==============================
The presence of kidney stones in individuals with type 2 diabetes mellitus (T2DM) poses a significant challenge and burden, yet the underlying pathogenesis remains elusive. This study aimed to reveal the relationship between the albumin-bilirubin (ALBI) score and kidney stones in adult patients with T2DM. This cross-sectional study was conducted using data from 9,511 eligible patients. The main outcome of interest was the incidence of kidney stones, with the ALBI score serving as the primary exposure factor. Logistic regression analysis was performed to calculate the odds ratios and 95% confidence intervals for the association between ALBI score and kidney stones. Our study found that a higher ALBI score was independently related to the presence of kidney stones in adult patients with T2DM. When the ALBI score was stratified into tertiles, compared to patients with an ALBI score in the T1 category, those with ALBI scores in the T2 and T3 categories exhibited a significantly higher prevalence of kidney stones after adjusting for multiple potential confounding factors. Additionally, our results revealed a non-linear relationship between ALBI score and the presence of kidney stones, which was further supported by subgroup and interaction analyses. These findings offer preliminary insights that could potentially inform future approaches to understanding kidney stone risk in adults with T2DM. Additional studies are needed to validate our conclusions.

Introduction

Kidney stone is a common and recurrent chronic disease, closely associated with factors such as diet, metabolism, genetics, and environment (Peerapen & Thongboonkerd, 2023). Type 2 diabetes mellitus (T2DM), a systemic metabolic disorder characterized by chronic hyperglycemia, predisposes patients to numerous complications such as cardiovascular diseases, peripheral neuropathy, retinopathy, diabetic complications due to prolonged metabolic derangements (Ruze et al., 2023). Numerous epidemiological investigations have demonstrated that individuals with diabetes exhibit a significantly elevated risk of both incident and recurrent nephrolithiasis compared to their non-diabetic counterparts (Aune et al., 2018; Lin et al., 2020; Ping et al., 2019). Notably, T2DM patients with concomitant kidney stones face heightened susceptibility to bacterial infections, which may exacerbate glycemic instability and perpetuate a vicious cycle of metabolic dysregulation and infectious complications (Wang et al., 2020; Wei et al., 2015). The substantial disease burden and healthcare expenditures associated with nephrolithiasis in this population underscore its critical public health implications. Despite these clinical challenges, the precise pathophysiological mechanisms and risk factors remain incompletely elucidated, necessitating further investigation into risk stratification strategies.

In recent years, the albumin-bilirubin (ALBI) score has emerged as a validated biomarker for hepatic functional assessment. Originally proposed by Johnson et al. (2015), it serves as an objective evidence-based grading system for hepatocellular carcinoma (HCC) patients (Johnson et al., 2015; Toyoda & Johnson, 2022). Apart from HCC, the clinical utility of the ALBI score has expanded to encompass prognostic evaluation in diverse non-hepatic pathologies (Shi, Zhang & Zhang, 2020; Yamada et al., 2021). However, no prior studies have systematically examined the potential association between ALBI score and nephrolithiasis risk in T2DM populations. Leveraging data from a large Chinese cohort, this study aims to preliminarily investigate the relationship between ALBI score and kidney stone occurrence in adults with T2DM.

Patients and Methods

Study design and population

To conduct this cross-sectional study, we screened 28,199 adult patients with T2DM from December 2015 to December 2023 at the First Affiliated Hospital of the University of Science and Technology of China. Subsequently, to avoid interfering with the reliability of the study results, we further excluded the following patients: (1) unrecorded baseline information; (2) loss of key laboratory results (such as bilirubin and albumin); (3) lack of urological imaging examination data; (4) female patients with pregnancy; (5) patients with malignancies; (6) severe renal function impairment with estimated glomerular filtration rate (eGFR) <30 mL/min/1.73 m2. Finally, we collected data from 9,511 eligible patients for this study (Fig. 1). Our study design was reviewed and approved by the ethics committee of the First Affiliated Hospital of USTC (No. 2023-RE-232). Informed consent was waived as all information was retrospectively extracted from medical records. All the data was anonymously processed and reported according to the Strengthening the Reporting of Observational Studies in Epidemiology (STROBE) guideline (Von Elm et al., 2007).

Figure 1 Patients’ selection flowchart.

(A) Patients with estimated glomerular filtration rate (eGFR) < 30 mL/min/1.73 m2. T2DM, type 2 diabetes mellitus.

Diagnosis of type 2 diabetes mellitus and kidney stones

Since the study population was from China, the Chinese guidelines for T2DM were referred to (Chinese Elderly Type 2 Diabetes Prevention and Treatment of Clinical Guidelines Writing Group et al., 2022; Jia et al., 2019). Patients met one of the following criteria were considered as T2DM: (1) typical symptoms of diabetes (polydipsia, polyuria, polyphagia, and weight loss) plus random plasma glucose ≥11.1 mmol/L. (2) Fasting plasma glucose ≥7.0 mmol/L. (3) Oral glucose tolerance test-2-hour postprandial plasma glucose ≥11.1 mmol/L. For patients without typical symptoms of diabetes, plasma glucose must be retested on a different day for confirmation. (4) Self-reported diagnosis of T2DM and taking oral medications or subcutaneous injection of insulin. All patients should be excluded from the diagnosis of type 1 diabetes mellitus and other specific types of diabetes. In this study, kidney stones were identified based on the report of ultrasonography and/or computed tomography of the urinary system, ‘kidney stone’ as a keyword for information extraction.

Collection of clinical and laboratory information

In this study, characteristics such as age, sex, height, weight, and marital status were directly accessed from the medical record. Body mass index (BMI) was calculated by weight (kg)/height (m)2. Diagnosis of hypertension was made by systolic blood pressure ≥140 mmHg and/or diastolic blood pressure ≥90 mmHg or a self-reported history and use of antihypertensive medication. Laboratory measurements were extracted from the Hospital Information System including glycated hemoglobin (HbA1c, %), red blood cell (RBC, ×1012/L), hemoglobin (Hb, g/L), white blood cell (WBC, ×109/L), platelet (PLT, ×109/L), alanine aminotransferase (ALT, IU/L), aspartate aminotransferase (AST, IU/L), alkaline phosphatase (ALP, IU/L), total bilirubin (TBiL, µmol/L), direct bilirubin (DBiL, µmol/L), indirect bilirubin (IBiL, µmol/L), serum total protein (STP, g/L), albumin (Alb, g/L), serum creatinine (Scr, µmol/L), blood urea nitrogen (BUN, mmol/L), uric acid (UA, µmol/L), total cholesterol (TC, mmol/L), triglyceride (TG, mmol/L), high-density lipoprotein (HDL, mmol/L), and low-density lipoprotein (LDL, mmol/L). The eGFR (mL/min/1.73 m2) was calculated by CKD-EPI creatinine (2021) (Inker et al., 2021). eGFR = 142 * minimum (Scr/κ, 1)α * maximum (Scr/κ, 1)−1.200 * 0.9938age * 1.012 (for females). Here, κ = 0.7 (females) or 0.9 (males); α = −0.241 (females) or −0.302 (males); Scr is in mg/dL (1 mg/dL = 88.4 µmol/L). The ALBI score was calculated by equation for the linear predictor as previous report (Johnson et al., 2015). ALBI score = (log10 bilirubin * 0.66) + (albumin * −0.085).

Outcome and exposure

The primary outcome of this study was the occurrence of kidney stones (yes or no), and the exposure factor for analysis was the ALBI score. The ALBI score is a continuous variable, which we also treated as a categorical variable. Initially, we were interested in the ALBI grades according to the criteria in the original literature (Grade 1, ALBI score ≤ −2.60; Grade 2, −2.60 <ALBI score ≤ −1.39; Grade 3, ALBI score >−1.39) (Johnson et al., 2015). Only 36 (0.4%) patients with ALBI grade 3, so we put ALBI grade 2 and ALBI grade 3 together in the analysis. However, the following analysis did not find significant differences between patients with ALBI grade 1 and ALBI grade 2–3 in terms of the incidence of kidney stones (Fig. 2). With reference to a common method in cross-sectional studies, we further divided the patients into three groups according to the tertiles of the ALBI score (T1, ALBI score ≤ −2.85; T2, −2.85 <ALBI score ≤ −2.55; T3, ALBI score >−2.55), with T1 serving as the reference group.

Figure 2 The incidence of kidney stones in patients with different ALBI grade and different ALBI score (group by tertiles).

ALBI, Albumin-Bilirubin.

Screening of covariates

Herein, we collected more than twenty potential variables. Sex (female or male), hypertension (yes or no) and marital status (never married, married or others) are categorical variables and others are continuous variables. For the purpose of selecting appropriate covariates, we performed univariate logistic analysis of each variable for predicting the presence of kidney stones. The results indicated that age, sex, RBC, Hb, ALT, ALP, ALBI score, eGFR, UA, TG, HDL, LDL were related risk factors of kidney stones (P < 0.05) (Table 1). Then, we excluded ALT, ALP (also the indicators related to liver function) and Hb (similar to RBC). Finally, we selected age, sex, RBC, eGFR, UA, TG, HDL, and LDL as the covariates in this study.

Table 1 Univariate and multivariate analysis for screening the risk factors of kidney stones.

Characteristics	Univariate analysis	Multivariate analysis	
	OR	95% CI	P value	OR	95% CI	P value	
Baseline information							
Age, years-old	0.988	0.984–0.993	<0.001	0.996	0.990–1.003	0.244	
Sex, n (%)							
Female	1	–	ref.	1	–	ref.	
Male	1.367	1.208–1.547	<0.001	1.203	1.045–1.384	0.010	
BMI, kg/m2	0.995	0.979–1.011	0.543				
Marital status, n (%)							
Never married	1	–	ref.				
Married	1.246	0.787–1.971	0.348				
Others	1.085	0.620–1.901	0.774				
Hypertension, n (%)							
No	1	–	ref.				
Yes	1.098	0.978–1.233	0.113				
Laboratory results							
HbA1c, %	1.019	0.990–1.049	0.203				
RBC, ×1012 /L	1.174	1.063–1.296	0.002	1.025	0.908–1.156	0.695	
Hb, g/L	1.006	1.003–1.009	0.001				
WBC, ×109/L	0.998	0.970–1.026	0.860				
PLT, ×109/L	1.000	0.999–1.001	0.908				
ALT, IU/L	0.996	0.993–0.999	0.014				
AST, IU/L	0.999	0.997–1.001	0.367				
ALP, IU/L	0.996	0.993–0.998	<0.001				
ALBI score	1.454	1.249–1.692	<0.001	1.578	1.341–1.857	<0.001	
Scr, µmol/L	1.001	0.999–1.003	0.441				
eGFR, mL/min/1.73 m2	1.004	1.001–1.006	0.015	1.004	1.00–1.008	0.033	
BUN, mmol/L	1.012	0.988–1.037	0.323				
UA, µmol/L	1.002	1.001–1.002	<0.001	1.002	1.001–1.002	<0.001	
TG, mmol/L	1.028	1.009–1.048	0.004	1.016	0.994–1.038	0.155	
HDL, mmol/L	0.766	0.624–0.940	0.011	0.835	0.672–1.039	0.105	
LDL, mmol/L	1.125	1.056–1.198	<0.001	1.110	1.034–1.191	0.004	
Notes.

BMI body mass index

HbA1c glycated hemoglobin A1c

RBC red blood cell

Hb hemoglobin

WBC white blood cell

PLT platelet

ALT alanine aminotransferase

AST aspartate aminotransferase

ALP alkaline phosphatase

ALBI Albumin-Bilirubin

Scr serum creatinine

eGFR estimated glomerular filtration rate

BUN blood urea nitrogen

UA uric acid

TG triglyceride

HDL high-density lipoprotein

LDL low-density lipoprotein

OR odds ratio

95%CI 95% confidence interval

ref reference

Statistical analyses

The normality of continuous variables was assessed using the Shapiro–Wilk tests. Non-normal distribution continuous variables were described as medians (interquartile ranges (IQR)) and compared using Mann–Whitney or Kruskal–Wallis tests. Categorical variables were presented as frequencies (percentages) and compared using χ2 tests. Binary logistic regression analysis was used to calculate odds ratios (OR) and 95% confidence intervals (CI) for the association between the ALBI score and kidney stones. Model 1 was adjusted for sociodemographic factors (age and sex). Model 2 was further adjusted for RBC, eGFR, UA, TG, HDL, and LDL. Trend tests were also conducted using the median value of each category of ABLI scores when it was treated as a categorical variable in logistic regression analysis. We then investigated the nonlinear dose–response relationship between ALBI score and kidney stones using restricted cubic splines with four knots, median ALBI score of T1 (−3.07) was taken as the reference point. We further explored the interactions between the ALBI score and all covariates in multivariate model 2 using likelihood ratio tests. Subgroup analyses were performed based on age (≤60 versus >60, years-old), sex (female versus male), RBC (≤4.4 versus >4.4, ×1012/L), eGFR (≤100.4 versus >100.4, mL/min/1.73m2), UA (≤318 versus >318, µmol/L), TG (≤1.5 versus >1.5, mmol/L), HDL (≤1.0 versus >1.0, mmol/L) and LDL (≤2.4 versus >2.4, mmol/L). All the variables were adjusted for age, sex, RBC, eGFR, UA, TG, HDL, and LDL except for the stratification variable itself. Statistical analyses were performed using IBM SPSS (version 25.0, Armonk, NY, USA) and R software (version 4.2.0) (http://www.R-project.org). Statistical significance was defined as the P value (two-tailed) <0.05.

Results

Characteristics of study population

After screening of 28,199 patients, we included data from 9,511 adult patients with T2DM to complete the following data analysis (Fig. 1). Of these patients, 1,334 (14.0%) patients had kidney stones, while 8,177 (86.0%) did not. The median (IQR) of age and BMI were 60 (53–70) years-old and 24.7 (22.6–27.0) kg/m2, respectively. 3,575 (37.6%) were female and 5,936 (62.4%) were male. Comparison results revealed that patients with kidney stones had elevated levels of TBiL and IBiL (P < 0.001), together with decreased levels of Alb (P < 0.001), indicative of compromised liver function. Subsequently, we computed the ALBI score for all participants, and further highlighting a significantly higher ALBI score among patients with kidney stones (P < 0.001).

Risk factors for kidney stones

Before exploring the association between ALBI score and kidney stones. We carried out univariate and multivariate analyses for preliminarily evaluating the relationship and meanwhile selecting appropriate adjusted covariates. Firstly, we conducted univariate analyses, and the results indicated a significant positive correlation between ALBI and kidney stones (OR=1.454, 95%CI [1.249–1.629], P < 0.001). Subsequently, the significant variables in the univariate analysis were incorporated into the multivariate model for adjustment, and the results still demonstrated that the ALBI score was a risk factor for kidney stones (OR=1.578, 95%CI [1.341–1.857], P < 0.001). For further analyses, we determined eight adjusted covariates with P < 0.05 in univariate analysis including two sociodemographic variables (age and sex) for adjustment in model 1 and six laboratory tests (RBC, eGFR, UA, TG, HDL, and LDL) for further adjustment in model 2 (Table 1).

Association between ALBI score and kidney stones

If we directly treated the ALBI score as a continuous variable, per unit increase in the ALBI score was associated with a 45.4%–57.8% higher risk of kidney stones in various models (Table 2). When transforming ALBI score into a categorical variable, we initially adopted the ALBI grade (grade 1, ALBI score ≤ −2.60; grade 2–3, ALBI score >−2.60) in light of previous research (Johnson et al., 2015). However, there was no significant difference in the incidence of kidney stones between patients classified as ALBI grade 1 and ALBI grade 2–3 (P = 0.117) (Fig. 2). The results of the multivariate analysis indicated a slightly elevated risk of kidney stones among patients with ALBI grade 2–3 (OR values ranging from 1.102 to 1.157 in different models) compared to patients with ALBI grade 1 (Table 2). This weak association might be attributable to population heterogeneity, given that the ALBI grade was originally developed for hepatocellular carcinoma patients. We subsequently converted the ALBI score into a categorical variable with tertiles defined as follows: T1 (ALBI score ≤ −2.85), T2 (−2.85 <ALBI score ≤ −2.55), and T3 (ALBI score >−2.55). Individuals with ALBI score T2 and T3 exhibited a significantly higher incidence of kidney stones compared to those with ALBI score T1 (P < 0.001) (Fig. 2). In comparison to patients with an ALBI score in the T1 category, those with an ALBI score in the T2 category had OR values ranging from 1.601 to 1.613, and patients with an ALBI score in the T3 category had OR values ranging from 1.366 to 1.452 in different models (Table 2).

Table 2 Associations between ALBI score and kidney stones in adult patients with type 2 diabetes mellitus.

	Case/total number	Crude model	Model 1 a	Model 2 b	
		OR (95%CI)	P value	OR (95% CI)	P value	OR (95% CI)	P value	
ALBI score	1,334/9,511	1.454 (1.249–1.692)	<0.001	1.525 (1.307–1.778)	<0.001	1.578 (1.341–1.857)	<0.001	
ALBI grade								
Grade 1 (≤ −2.60)	829/6,095	1	ref.	1	ref.	1	ref.	
Grade 2–3 (>−2.60)	505/3,416	1.102 (0.978–1.242)	0.111	1.148 (1.017–1.295)	0.025	1.157 (1.022–1.310)	0.022	
ALBI score tertiles								
T1 (≤ −2.85)	346/3,141	1	ref.	1	ref.	1	ref.	
T2 (>−2.85 and ≤ −2.55)	531/3,210	1.601 (1.384–1.852)	<0.001	1.607 (1.389–1.859)	<0.001	1.613 (1.393–1.869)	<0.001	
T3 (>−2.55)	457/3,160	1.366 (1.176–1.586)	<0.001	1.430 (1.230–1.663)	<0.001	1.452 (1.243–1.696)	<0.001	
Trent test			<0.001		<0.001		<0.001	
Notes.

ALBI Albumin-Bilirubin

OR odds ratio

95%CI 95% confidence interval

ref reference

a Adjusted for age and sex.

b Adjusted for model 1 + red blood cell (RBC), estimated glomerular filtration rate (eGFR), uric acid (UA), triglyceride (TG), high-density lipoprotein (HDL) and low-density lipoprotein (LDL).

Nonlinearity, subgroup and interaction analyses

The study further validated the nonlinear relationship between ALBI score and kidney stones through the utilization of restricted cubic splines in various models, all of which demonstrated statistical significance for non-linearity (P < 0.001) (Fig. 3). In subgroup analyses, variables were stratified based on the median values of age, RBC, eGFR, UA, TG, HDL and LDL. Figure 4 showed the results of the subgroup and interaction analyses. A significant positive correlation was observed across various subgroups between the tertiles of the ALBI score and the presence of kidney stones in addition to individuals with age ≤60 years-old and RBC level >4.4 ×1012/L. No significant interactions were noted in relation to age, sex, RBC, eGFR, UA, HDL, and LDL. Despite a P-value of 0.027 in the interaction analysis between the ALBI score and TG, a positive association persisted and multiple comparison factors should be taken into account.

Discussion

The ALBI score was originally developed as a measure of liver function in patients with HCC (Johnson et al., 2015). Since the publication of the ALBI score, numerous studies have demonstrated its correlation with cancers except HCC, including pancreatic cancer, colorectal cancer, gastric cancer, cholangiocarcinoma, and even brain cancer (Abdel-Rahman, 2019; Li et al., 2021; Ni et al., 2019; Watanabe et al., 2021). Ni et al. (2019) demonstrated ALBI score can effectively predict the long-term prognosis of patients with intrahepatic cholangiocarcinoma undergoing CT-guided percutaneous microwave ablation, with cumulative overall survival rates at 1, 3, and 5 years being 89.5%, 52.2%, and 35.0%, respectively. Although most studies attempt to link the ALBI score with specific cancers and liver function, we should consider whether the association between the ALBI score and mortality is truly present due to its impact on aspects of liver dysfunction. Alternatively, it could simply reflect the prognostic capabilities of serum albumin and bilirubin levels, independent of liver function. Previous studies have discussed the role of albumin as an indicator of malnutrition, as well as the immunomodulatory and anti-inflammatory effects of albumin and bilirubin (Deng et al., 2022; Ieda et al., 2022; Zhang et al., 2023). This further supports the relative non-specificity of the ALBI score and its broad applicability. Additionally, some studies have confirmed the correlation between the ALBI score and non-hepatic diseases, including acute or chronic heart failure, acute pancreatitis, and aortic dissection (Liu et al., 2021; Shi, Zhang & Zhang, 2020; Yamada et al., 2021).

ALBI score has been being widely used by healthcare specialists, this is perhaps due to its objectivity and simplicity. However, no studies have yet explored the relationship between the ALBI score and the incidence of kidney stones in patients with T2DM. This study primarily found that, after adjusting for potential confounding factors, an increase in the ALBI score is associated with a higher risk of kidney stones. ALBI is a relatively independent risk factor for kidney stones in T2DM patients. The association remained robust in the subgroup and interaction analyses. This is consistent with previous findings regarding the ALBI score in chronic non-liver diseases, indicating its non-specific nature. The mechanism of kidney stones in patients with T2DM remains unclear, but numerous studies suggest that both conditions are strongly associated with inflammation (Di et al., 2023; Guo et al., 2022; Khan, Canales & Dominguez-Gutierrez, 2021; Sas et al., 2022). Inflammation and oxidative stress may play important roles and could explain this result, but further in-depth studies are needed to validate this hypothesis.

Figure 3 The restricted cubic splines displayed the nonlinear association between ALBI score and kidney stones in adult patients with type 2 diabetes mellitus.

(A) Crude model; (B) Model 1, adjusted for age and sex; (C) Model 2, adjusted for model 1 + red blood cell (RBC), estimated glomerular filtration rate (eGFR), uric acid (UA), triglyceride (TG), high-density lipoprotein (HDL) and low-density lipoprotein (LDL). ALBI, Albumin-Bilirubin; OR, odds ratio; 95%CI, 95% confidence interval.

Figure 4 Associations between ALBI score and kidney stones in different subgroups.

All the variables were adjusted for age, sex, RBC, eGFR, UA, TG, HDL, LDL except for the stratification variable itself. ALBI, Albumin-Bilirubin; RBC, red blood cell; eGFR, estimated glomerular filtration rate; UA, uric acid; TG, triglyceride; HDL, high-density lipoprotein; LDL, low-density lipoprotein; OR, odds ratio; 95% CI, 95% confidence interval.

Kidney stones are one of the most frequent and common diseases in urology, with complex causes and high recurrence rates. Many studies have found that diabetic patients have a higher risk of kidney stone occurrence and recurrence compared to non-diabetic patients. In a retrospective analysis by Li et al. (2019), involving 297 patients with asymptomatic kidney stones with an average follow-up of 4.2 years, it was discovered that patients with diabetes are more prone to kidney stone formation, with a higher risk of stone-related complications. In a study conducted by Prasanchaimontri & Monga (2020) on the recurrence of kidney stones in patients with T2DM, a total of 1,617 T2DM patients who underwent kidney stone surgery were included. Among them, 373 patients (23%) experienced stone recurrence, with a median time to recurrence of 64 months. During the treatment of kidney stones, patients with T2DM who also have kidney stones may face increased risks of postoperative infections and complications. Therefore, investigating the risk factors for kidney stones in T2DM patients is crucial, as it can provide theoretical support for clinicians managing such patients. This study first discovered the relationship between the ALBI score and the incidence of kidney stones in adult T2DM patients. The ALBI score may assist clinicians in early identification of high-risk patients with T2DM complicated by kidney stones, enabling timely adjustments to treatment plans and potentially preventing the onset of kidney stones. For instance, evaluating the ALBI score in T2DM patients prior to kidney stone intervention and concurrently implementing strategies to optimize liver function may potentially improve surgical outcomes. In the future, more studies should be conducted to further validate the application of the ALBI score in different clinical scenarios, particularly its effectiveness in guiding specific clinical interventions.

However, the present study has some limitations that need to be acknowledged. First, due to the cross-sectional nature of our analysis, causality cannot be inferred from the results. Given the complexity of this association, additional longitudinal studies and investigations may reveal the potential of the ALBI score in predicting the clinical outcomes of kidney stones in T2DM patients. Second, some important information was missed in this study, such as stone compositions, course of T2DM and laboratory indicators of urine. Third, this study was carried out in a Chinese population; more external validations are needed to confirm whether our findings can be generalized to other populations. Finally, despite the construction of a multivariate logistic regression model and performing subgroup analyses to control for potential confounders that influence the relationship between the ALBI score and kidney stones, residual confounding effects cannot be completely ruled out. Therefore, future multicenter cohort studies will be necessary to corroborate these findings, incorporating more potential confounders and utilizing standardized and consistent measures.

Conclusions

In this cross-sectional study, we observed a potential association between the ALBI score and kidney stone incidence in adult patients with T2DM. Our findings offer potential implications for the management and prevention of kidney stones in individuals with T2DM; however, further evaluation and validation of these results are warranted in future research endeavors.

Supplemental Information

Supplemental Information 1 Raw data

Supplemental Information 2 STROBE checklist

Additional Information and Declarations

Competing Interests

Author Contributions

Human Ethics

Ethics

Data Availability

The authors declare there are no competing interests.

Wenya Mo performed the experiments, analyzed the data, prepared figures and/or tables, authored or reviewed drafts of the article, and approved the final draft.

Tao Zhou performed the experiments, prepared figures and/or tables, data acquisition, and approved the final draft.

Qifei Dong performed the experiments, prepared figures and/or tables, data acquisition, and approved the final draft.

Yuhan Chen performed the experiments, prepared figures and/or tables, data acquisition, and approved the final draft.

Xuechun Hu performed the experiments, analyzed the data, prepared figures and/or tables, and approved the final draft.

Jun Xiao conceived and designed the experiments, authored or reviewed drafts of the article, and approved the final draft.

Changming Wang conceived and designed the experiments, analyzed the data, prepared figures and/or tables, authored or reviewed drafts of the article, and approved the final draft.

Ping Xiang conceived and designed the experiments, prepared figures and/or tables, authored or reviewed drafts of the article, and approved the final draft.

The following information was supplied relating to ethical approvals (i.e., approving body and any reference numbers):

The ethics committee of the First Affiliated Hospital of USTC (No. 2023-RE-232).

The following information was supplied relating to ethical approvals (i.e., approving body and any reference numbers):

The ethics committee of the First Affiliated Hospital of USTC (No. 2023-RE-232).

The following information was supplied regarding data availability:

The raw data is available in the Supplementary File.

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
