# Peer review of "Elevated albumin-bilirubin score as a predictor of kidney stones in adults with type 2 diabetes mellitus: evidence from a cross-sectional study"

_PeerJ, doi:10.7717/peerj.19419_

## Round 0.1 · original submission · Major Revisions

The presentation of the results, explaining the statistical methods and the Discussions should be expanded and more clearly systematized.

Reviewer 1 ·

Basic reporting

Thank you to the authors for this well-written manuscript. This cross-sectional study addresses a significant issue relevant to the daily practice of urologists.

• Language and Clarity: The manuscript is written in professional and clear English. However, minor grammatical and phrasing issues can be improved for better readability, such as in the introduction.
• Background and References: The background information is thorough, with relevant references provided to establish context. Literature citations is up to date appropriately reflect the field’s current understanding.
• Structure and Presentation: The manuscript follows a standard structure with a clear flow between sections. Figures and tables are relevant, well-labeled, and described in the text.
• Raw Data: Raw data availability is noted, and essential details are shared in compliance with the journal’s policies.

Experimental design

Original Research: The study addresses a novel question by linking the albumin-bilirubin (ALBI) score with kidney stone incidence in T2DM patients, aligning well with the journal’s scope.
• Research Question and Gap: The research question is well-defined and fills a meaningful gap in understanding the predictive utility of the ALBI score for kidney stones in a diabetic population.
• Methodology: The cross-sectional study design is described in detail. The inclusion and exclusion criteria, data collection process, and statistical methods are clearly outlined and justify the rigor of the investigation.
• Ethical Standards: Ethical approval is obtained, and the study adheres to the Declaration of Helsinki. Retrospective data collection ensures no direct patient involvement, minimizing ethical concerns.
• Replicability: Methods are sufficiently detailed to allow replication, including equations for calculating ALBI scores and statistical analyses.

Validity of the findings

• Data Robustness: The analysis is robust, with appropriate statistical methods used to adjust for confounders. Logistic regression and restricted cubic spline analyses effectively address non-linearity and interactions.
• Conclusions: Conclusions are supported by the data and appropriately linked to the study’s aims. However, claims about causality are overstated given the cross-sectional design.
• Limitations: The study acknowledges limitations such as the cross-sectional nature, lack of external validation, and absence of detailed stone composition data. These are reasonable given the study’s scope but warrant further research.

Additional comments

The study’s strength lies in its large sample size, rigorous statistical adjustments, and novel focus on the ALBI score. Subgroup analyses add depth to the findings. However, they can improve the discussion section, expanding the clinical applicability of findings, particularly how ALBI score monitoring could guide interventions.

Reviewer 2 ·

Basic reporting

No comments

Experimental design

The text lacks a smooth narrative flow. For instance, the exclusion criteria and patient selection process are heavily detailed without an engaging transition into the study's rationale.
The rationale for some methodological decisions (e.g., why certain exclusion criteria were chosen or why ALBI grades were combined) is not fully explained.
Terms like "uncertainly diagnosis of kidney stones" and "loss of key laboratory results" are vague and lack precision.
The text often cites previous guidelines or studies (e.g., "with reference to the guideline of T2DM in China and previous studies") without summarizing their relevance or integrating them into the narrative.

Validity of the findings

The analytical strategy for the ALBI score is underexplained. The decision to divide the score into tertiles and the lack of significant findings between grades are presented without sufficient context or discussion. Include a brief discussion of why tertiles were used and the implications of the lack of significant differences.
The results are presented in a fragmented and non-linear manner. For example, the transition from univariate to multivariate analysis lacks coherence, and the nonlinear relationship is discussed abruptly. The reporting of odds ratios (ORs) and confidence intervals (CIs) is exhaustive but overwhelming, particularly when presented in multiple models and categories. Summarize key findings in one or two sentences and refer readers to a comprehensive table for detailed ORs and CIs.

---

## Round 0.2 · accepted · Accept

The authors have adequately resolved all the reviewers comments. No furthe suggestions.

Reviewer 2 ·

Basic reporting

No comment

Experimental design

No comment

Validity of the findings

No comment

Additional comments

Dear authors, after the major revision, I think the text is generally well-structured, clear, and scientifically sound. The article is usually well-structured, clear, and scientifically sound. In my opinion, the editorial process can continue.